# Triterpene Derivatives from *Garcinia oligantha* and Their Anti-Cancer Activity

**DOI:** 10.3390/plants12010192

**Published:** 2023-01-03

**Authors:** Xiaohui Peng, Chao Wang, Yonglian Hou, Jiamei Tian, Xiaojie Fan, Dahong Li, Huiming Hua

**Affiliations:** Key Laboratory of Structure-Based Drug Design & Discovery, Ministry of Education, School of Traditional Chinese Materia Medica, Shenyang Pharmaceutical University, Shenyang 110016, China

**Keywords:** *Garcinia oligantha*, triterpene, anti-cancer

## Abstract

Phytochemical investigations of leaves and twigs from *Garcinia oligantha* Merr. resulted in the isolation of five undescribed triterpene derivatives (**1**–**5**) and six known analogs (**6**–**11**). Their structures were determined based on extensive spectroscopic data and high-resolution mass spectra analyses. Compounds **1**–**11** were tested for their in vitro cytotoxicity against three human cancer cell lines (HeLa, HepG-2, and MCF-7). Compounds **1**, **2**, **8,** and **11** exhibited broad and significant cytotoxicity against the tested cell lines with IC_50_ values ranging from 5.04 to 21.55 μM. Compounds **5** and **9** showed cytotoxicity against HeLa and MCF-7 with IC_50_ values ranging from 13.22 to 19.62 μM. The preliminary structure–activity relationship for the 11 isolated compounds is also discussed.

## 1. Introduction

The genus *Garcinia*, belonging to the family Clusiaceae or Guttiferae, has a rich source of compounds, including phloroglucinols [1,2,3], xanthones [4,5,6,7,8,9,10,11,12,13], and biflavonoids [14,15,16]. Dauphinol B, a phloroglucinol isolated from *G. dauphinensis*, showed potent antiplasmodial activity against the Dd2 drug-resistant strain of *Plasmodium falciparum*, with an IC_50_ of 0.8 μM [17]. 7-Epiclusianone from *G. brasiliensis* showed important biological effects, including anti-cancer [18], anti-inflammatory [19,20], and antianaphylactic activities [21]. *α*-Mangostin, a xanthone from *G. mangostana*, showed anti-biofilm [22,23], antimicrobial [24], and antidiabetic effects [25]. Kolaviron, a biflavonoid from *G. kola*, has antimalarial activities in *P. berghei*-infected mice [26]. In the last 20 years, our group’s focus on investigating *Garcinia* plants led to the isolation and characterization of many bioactive compounds, including depsidones, xanthones, biflavonoids, triterpenes, and biphenyl derivatives from *G. paucinervis* [27,28,29,30], *G. bracteata* [31,32,33], *G. lancilimba* [34], *G. nujiangensis* [35], *G. multiflora* [36], *G. xanthochymus* [37], and *G. hanburyi* [38,39].

*G. oligantha*, a 1–3 m tall shrub, has been used as a traditional Chinese herbal medicine to treat fevers, toothaches, and scalds [40]. Previous phytochemical investigations of the plant led to the isolation of many xanthone derivatives with anti-cancer and/or anti-inflammatory activities [4,10,41,42], but triterpenoids were rarely reported from it.

In our continuing search for biologically active and structurally unique compounds from the genus *Garcinia*, we investigated the chemical constituents of *G. oligantha*. Present phytochemical studies on the acetone extract of the leaves and twigs of *G. oligantha* afforded 11 triterpene derivatives (Figure 1), including five previously undescribed natural products, 2*α*,3*β*-diacetyl arjunolic acid (**1**), 3*β*,23-diacetyl arjunolic acid (**2**), 2*α*-acetyl arjunolic acid (**3**), 23-acetyl hovenic acid (**4**), and 2*α*,23-diacetyl hovenic acid (**5**), and six known triterpenoids, viz., betulinic acid (**6**) [43,44], betulin (**7**) [45], 3-hydroxy-23-acetoxy-lup-20(29)-en-28-oic acid (**8**) [46], 23-hydroxybetulinic acid (**9**) [47], tomentoid B (**10**) [48], and 2*α*,23-diacetoxy-3*β*-hydroxyolean-12-en-28-oic acid (**11**) [49,50]. These products’ in vitro cytotoxicities against three human cancer cell lines, HeLa (human cervical cancer), HepG-2 (human hepatocellular carcinoma), and MCF-7 (human breast cancer), were evaluated. In this paper, the isolation, structure elucidation of all compounds, and anti-cancer activities of **1**–**11** are reported.

## 2. Results

### 2.1. Isolation and Structure Elucidation of Triterpenoids

The dried leaves and twigs of *G. oligantha* were extracted with acetone (30 L × 72 h × 3) at room temperature. After removing the acetone under reduced pressure, the residue was extracted sequentially with petroleum ether (PE), dichloromethane (DCM), and methanol. The PE-soluble and DCM-soluble fractions were separated by silica gel, Sephadex LH-20 column chromatography, and semipreparative HPLC and purified by crystallization to yield 11 triterpene derivatives.

The molecular formula of compound **1** was determined as C_34_H_52_O_7_ based on the HRESIMS data at *m/z* 571.3633 [M − H]^-^ (calcd 571.3640). The ^1^H NMR spectrum (Table 1) of **1** displayed signals corresponding to six tertiary methyls at *δ*_H_ 1.13 (3H, s), 1.08 (3H, s), 0.92 (3H, s), 0.90 (3H, s), 0.74 (3H, s), and 0.72 (3H, s), and two acetyl groups at *δ*_H_ 2.08 (3H, s) and 2.00 (3H, s), as well as one olefinic proton at *δ*_H_ 5.26 (1H, t, *J* = 3.8 Hz). The ^13^C NMR spectrum (Table 2) of **1** showed 34 carbon signals assigned to eight primary, ten secondary, six tertiary, and ten quaternary carbons, including two acetyl groups and the typical olefinic signals at *δ*_C_ 144.0 and 122.2 for the double bond at C-12 (13) of an oleanane-type triterpene. The ^1^H and ^13^C NMR spectra of **1** were similar to those of arjunolic acid except for the presence of two acetyl groups instead of two hydroxy groups in arjunolic acid [51]. The HMBC spectrum showed cross-peaks from H-2 and H_3_-32 to C-31 and from H-3 and H_3_-34 to C-33, confirming two acetyl groups located at C-2 and C-3, respectively (Figure 2). The stereochemistry of **1** was determined by analyzing the coupling constants and the NOESY experiment. The coupling constant (*J* = 10.7 Hz) between H-2 and H-3 indicated the orientation of two protons possessing dual-axial bonds. The NOESY spectrum showed the cross-peaks of H-2 with H_3_-24, H_3_-24 with H_3_-25, H_3_-25 with H_3_-26, and H-18 with H_3_-30, indicating the *β* orientation in these protons. The NOESY correlations of H-3 with H-5 and H_2_-23 and H-9 with H-5 and H_3_-27 suggested their *α*-orientation (Figure 3). After the hydrolysis of **1**, compound **1a** was obtained, and its 1D NMR data and specific rotation were the same as those of arjunolic acid, which further confirmed the above conclusion [51,52]. Thus, the structure of **1** was determined, as shown in Figure 1, and named 2*α*,3*β*-diacetyl arjunolic acid.

Compound **2** displayed an [M − H]^−^ ion peak at *m/z* 571.3639 (calcd 571.3640) in its negative mode HRESIMS, corresponding to the molecular formula C_34_H_52_O_7_. The ^1^H NMR data (Table 1) showed eight methyls at *δ*_H_ 2.11 (3H, s), 2.08 (3H, s), 1.12 (3H, s), 1.01 (3H, s), 0.92 (3H, s), 0.90 (3H, s), 0.84 (3H, s), and 0.74 (3H, s), and one olefinic proton at *δ*_H_ 5.28 (1H, t, *J* = 3.4 Hz). The ^13^C NMR spectrum (Table 2) showed 34 carbon signals, including three carbonyls at *δ*_C_ 183.87, 172.32, and 171.03, and two olefinic carbons at *δ*_C_ 143.8 and 122.4. The ^13^C and ^1^H NMR data of **1** and **2** are very similar, with major differences being the chemical shifts of C-2, and H-2 shifted upfield from *δ*_C_ 69.6, and *δ*_H_ 5.23 to *δ*_C_ 67.6, and *δ*_H_ 3.86; C-23 and H_2_-23 were shifted downfield from *δ*_C_ 64.6, and *δ*_H_ 3.37, 2.89 to *δ*_C_ 65.5, and *δ*_H_ 3.82, 3.65, respectively, suggesting that the acetoxy at C-2 in **1** was instead located at C-23 in **2**. This finding was further supported by the HMBC correlation from H_2_-23 to C-31. We determined the relative configuration of **2** according to the NOESY correlations (Figure 3) and the coupling constant (10.1 Hz) between H-2 and H-3. Hence, the structure of **2** was confirmed, as shown in Figure 1, and named 3*β*,23-diacetyl arjunolic acid.

Compound **3** was isolated as a white amorphous powder. The pseudomolecular ion peak at *m/z* 531.3694 [M + H]^+^ in its HRESIMS suggested C_32_H_50_O_6_ as its molecular formula. The ^1^H NMR (Table 1) of **3** revealed the presence of seven methyls at *δ*_H_ 2.05 (3H, s), 1.17 (3H, s), 1.08 (3H, s), 0.94 (3H, s), 0.90 (3H, s), 0.84 (3H, s), and 0.73 (3H, s), and one olefinic proton at *δ*_H_ 5.23 (1H, br s). The ^13^C NMR spectrum (Table 2) showed 31 signals, including a carbonyl at *δ*_C_ 173.0 and two olefinic carbons at *δ*_C_ 145.7 and 123.1, respectively. Comparing the 1D NMR data of **3** to those of tomentoid B (**10**) [48] indicated that the two compounds were closely related. The obvious spectroscopic difference between them resulted from the presence of the acetoxy group at C-2 in **3**, instead of C-23 in **10**. The shielded chemical shifts of C-23 and H_2_-23 of **3** further supported the above assignment. Its similar NMR data and same molecular formula as tomentoid B (**10**) indicated that **3** also had a carboxyl group at C-17, of which the carbon signal was missing. Fortunately, the carboxyl carbon signal was observed at 181.1 in the ^13^C NMR spectrum measured in CDCl_3_ (Appendix A). The large proton spin-coupling constant (*J* = 10.8 Hz) of H-3 with H-2 suggests that the protons at C-2 and C-3 are trans-axial. Furthermore, the relative stereochemistry of **3** was established by the NOESY cross-peaks shown in Figure 3. Therefore, the structure of **3** was named 2*α*-acetyl arjunolic acid (Figure 1).

Compound **4** was obtained as a white amorphous powder. Its molecular formula was deduced as C_32_H_50_O_6_ based on a negative-ion at *m/z* 529.3528 [M − H]^−^ (calcd for C_32_H_49_O_6_^−^, 529.3535). Signals of six methyl singlets at *δ*_H_ 2.06 (3H, s), 1.70 (3H, s), 1.01 (3H, s), 0.97 (3H, s), 0.96 (3H, s), 0.75 (3H, s), and two olefinic protons at *δ*_H_ 4.72 (1H, d, *J* = 2.2 Hz) and 4.60 (1H, br s) were observed in the ^1^H NMR spectrum (Table 1). The ^13^C NMR spectrum (Table 2) showed 32 carbon signals, including six methyls, two carbonyls, and two olefinic carbons. The ^1^H and ^13^C NMR spectra of **4** were similar to those of the known compound hovenic acid [53], with the most noticeable difference observed for the hydroxy at C-23 in hovenic acid replaced by an acetoxy in **4**. The HMBC correlations from H_2_-23 and H_3_-32 to C-31 confirmed the above deduction. The stereochemistry of **4** was determined by analyzing the coupling constants and NOESY data. The large spin-coupling constant (*J*_H-2, H-3_ = 9.7 Hz) indicated that the 2,3-dihydroxyl groups should have a 2*α*,3*β*-orientation, further supported by the NOESY correlations of H-2 with H_3_-24, and H-3 with H_2_-23 (Figure 3). We can confirm the above conclusion by comparing the compound’s specific rotation and NOESY spectrum to hovenic acid [53,54]. Thus, we determined the structure of **4** as 23-acetyl hovenic acid (Figure 1).

Compound **5** was obtained as a white amorphous powder. Its molecular formula was determined as C_34_H_52_O_7_ based on the HRESIMS. The ^1^H NMR spectrum (Table 1) of **5** in CD_3_OD (600 MHz) displayed signals corresponding to five tertiary methyls at *δ*_H_ 1.70 (3H, s), 1.01 (6H, s), 0.97 (3H, s), 0.80 (3H, s), two acetoxy groups at *δ*_H_ 2.07 (3H, s), 2.04 (3H, s), oxygenated methylene at *δ*_H_ 3.95 (2H, s), and two olefinic protons at *δ*_H_ 4.71 (1H, s) and 4.59 (1H, s), respectively. The ^13^C NMR spectrum (Table 2) of **5** revealed signals corresponding to 34 carbon atoms, including seven methyls, three carbonyls, and a double bond. The ^1^H and ^13^C NMR data of **5** were identical to those of **4**, except for the existence of one more acetoxyl group in **5** at *δ*_H_ 2.04 (3H, s) and *δ*_C_ 172.9 and 21.3. Compared to the ^1^H NMR spectrum of **4**, the resonance of H-2 shifted downfield from *δ*_H_ 3.68 to 5.01, and the HMBC correlation of H-2 to C-33 in **5** suggested the acetoxy group located at C-2. The coupling constant (*J*_H-2, H-3_ = 10.1 Hz) established the 2*α*,3*β*-orientation of 2-OH and 3-OH, further confirmed by the NOESY correlations (Figure 3) of H-2 with H_3_-24 and H-3 with H_2_-23. Therefore, we determined the structure of **5** as 2*α*,23-diacetyl hovenic acid.

### 2.2. Evaluation of Biological Activity of Compounds **1**–**11**

We tested the anti-tumor activities of compounds **1**–**11** against three cancer cell lines (HeLa, HepG-2, and MCF-7).

Compounds **1**–**3**, **10,** and **11** are oleanane-type triterpenoids. Compounds **1**, **2,** and **11** with two acetoxys had stronger inhibitory activity than compounds **3** and **10** with one acetoxy, indicating that the more acetylated the hydroxyl groups, the stronger the activity.

Compounds **4**–**9** belong to lupane-type triterpenoids. Among them, compound **8** was the strongest inhibitor with IC_50_ values of 5.04–9.76 μM, whereas compounds **6** and **7** exhibited no cytotoxicity to the three cell lines, indicating that 23-hydroxyl or 23-acetoxy group can increase inhibitory activity. Compound **4** exhibited moderate activity against HeLa and HepG-2 cell lines, while **5** and **9** had moderate activity against HeLa and MCF-7 cell lines (Table 3).

## 3. Discussion

In previous phytochemical investigations on the genus *Garcinia*, pentacyclic triterpenoids were rarely reported [55]. In this study, we isolated 11 pentacyclic triterpenoids, including five new natural products (**1**–**5**), from the leaves and twigs of *G. oligantha.* They belong to oleanane- and lupane-type triterpenes, most of which possess 23-acetoxyl or acetoxy groups. Compounds **1**–**11** were tested for their cytotoxic activity against HeLa, HepG-2, and MCF-7 cell lines. Compound **8** showed the highest anti-cancer activity against these three human cancer cell lines, with IC_50_ values ranging from 5.04 to 9.76 μM. Among the five oleanane-type triterpenoids, the derivatives with two acetoxy groups showed more potent activity than those with one acetoxy group. For lupane-type triterpenes, the ones possessing 23-hydroxyl or 23-acetoxy groups were more active. The above results may support future investigations into the anti-tumor drug design of triterpenoids.

## 4. Materials and Methods

### 4.1. General Experimental Procedure

The UV spectra were obtained by a Shimadzu UV-2600i spectrometer (Shimadzu, Kyoto, Japan). Optical rotations were measured by a JASCO P-2000 polarimeter (Anton Paar, Ostfildern, Germany). We recorded the HRESIMS data on Bruker microTOFQ-Q mass spectrometers (Billerica, MA, USA) and an Agilent 6550 Q-TOF (Agilent Technologies, Palo Alto, CA, USA). The NMR spectra were recorded on a Bruker AVANCE Ⅲ HD 600 MHz NMR spectrometer (Bruker BioSpin, Billerica, MA, USA) using TMS as an internal standard. The chromatographic silica gel (Qingdao Haiyang Chemical Factory, Qingdao, China), ODS (YMC Co., Ltd., Kyoto, Japan), and Sephadex LH-20 (GE Healthcare, Uppsala, Sweden). We recorded the analytical HPLC data using a Shimadzu SPD-M20A series machine equipped with a YMC C-18 column (250 mm × 4.6 mm, 5 μm). We conducted the semipreparative HPLC using a Shimadzu LC-6AD series pumping system equipped with an SPD-20A UV detector and C-18 column (20 mm × 250 mm, 5 μm; YMC Co., Ltd.). All the organic solvents were purchased from Yuwang and Laibo Chemicals Industries, Ltd., Shenyang, China.

### 4.2. Plant Material

The plant material was purchased from Kunming Plant Classification Biotechnology Co., Ltd., and collected from Diaoluo Mountain (GPS coordinates 109°41′38″~110°4′46″ E, 18°38′42″~18°50′22″ N), Lingshui County, Hainan Province, China, in March 2019, which was authenticated as *G. oligantha* by Mr. Jun Zhang of Kunming Plant Classification Biotechnology Co., Ltd. Kunming, China. A voucher specimen (DHSZZ-201903) was deposited in the Department of Natural Products Chemistry, Shenyang Pharmaceutical University, Shenyang, China.

### 4.3. Extraction and Isolation

The air-dried leaves and twigs of *G. oligantha* (3.07 kg) were extracted with acetone (3 × 30 L) at room temperature to produce 387 g of dried resin, which was then extracted by PE, DCM, and methanol, successively.

The PE-soluble part (50.2 g) was fractionated by column chromatography (CC) over silica gel using solvent mixtures of PE-acetone (100:0–0:100, *v*/*v*) to obtain nine fractions (P1–P9). Fraction P3 (1.9 g) was separated over a silica gel column and eluted with CH_2_Cl_2_-MeOH (0:1–1:0) to yield subfractions P3.1–P3.7. Fr. P3.3 (736.3 mg) was separated over a silica gel column and eluted with PE-DCM (0:1–1:0, *v*/*v*) to produce ten subfractions P3.3.1–P3.3.7. Fr. P3.3.6 (23.6 mg) was purified by Sephadex LH-20 (MeOH) and then isolated by semipreparative HPLC (93% MeOH-H_2_O) to yield **12** (*t*_R_ = 29.9 min, 6.5 mg).

The DCM-soluble part (128.0 g) was subjected to silica gel CC with a PE-EtOAc gradient system to produce six fractions (D1–D6). Fraction D2 (3107.2 mg) was chromatographed on ODS with a MeOH-H_2_O gradient system to produce six subfractions (D2.1–D2.6). Fr. D2.4 (2214.4 mg) was subjected to a Sephadex LH-20 column (MeOH) to produce subfractions D2.4.1–D2.4.5. Fr. D2.4.3 (1167.7 mg) was further purified by semipreparative HPLC using a solvent of 80% MeOH-H_2_O from 0 to 100 min, and ending with 100% MeOH from 100 to 120 min. This process created three subfractions D2.4.3.1 (224.1 mg), D2.4.3.2 (195.2 mg), and D2.4.3.3 (543.9 mg). Fr. D2.4.3.2 and D2.4.3.3 were repeatedly recrystallized from MeOH to obtain **6** (10.3 mg) and **7** (8.9 mg), respectively.

Fraction D4 (23.0 g) was purified by CC over ODS using solvent mixtures of MeOH-H_2_O (60:40–100:0, *v*/*v*) to produce five subfractions D4.1–D4.5. Fr. D4.3 was separated by a silica gel CC (PE/acetone) to obtain four subfractions D4.3.1–D4.3.4. Fr. D4.3.1 (1359.3 mg) was separated by Sephadex LH-20 eluted with MeOH to produce three subfractions D4.3.1.1–D4.3.1.3. Fr. D4.3.1.1 (1000.5 mg) was purified by semipreparative HPLC, eluted with 84% MeOH-H_2_O, and further separated by semipreparative HPLC (80% MeOH-H_2_O) to yield **8** (*t*_R_ = 45.8 min, 7.2 mg).

Fraction D.5 (12.0 g) was chromatographed on an ODS CC and eluted with MeOH-H_2_O (55:45 to 100:0) in a step-gradient manner to produce four subfractions D5.1–D5.4. Fr. D5.2 (2308.5 mg) was separated by Sephadex LH-20 with MeOH to produce four subfractions D5.2.1–D5.2.4. Fr. D5.2.2 (183.4 mg) was subjected to a semipreparative HPLC (75% MeOH-H_2_O) to obtain three subfractions (D5.2.2.1–D5.2.2.3). Fr. D5.2.2.2 (14.4 mg) was purified by semipreparative HPLC (60% CH_3_CN-H_2_O) to obtain **4** (*t*_R_ = 47.2 min, 5.3 mg). Subfraction D5.2.2.3 was separated by the same method to obtain **3** (*t*_R_ = 48.5 min, 1.8 mg) and **10** (*t*_R_ = 50.7 min, 3.1 mg). Fr. D5.3 (2042.7 mg) was separated over Sephadex LH-20 (MeOH), and then purified by semipreparative HPLC (83% MeOH-H_2_O, from 0 to 40 min; 100% MeOH, from 40 to 60 min) to yield **2** (*t*_R_ = 17.1 min, 8.4 mg) and **11** (*t*_R_ = 25.2 min, 40.2 mg) and subfraction D5.3.3.3 (*t*_R_ = 40.0–60.0 min, 155.9 mg). Fr. D5.3.3.3 was subjected to semipreparative HPLC (76% MeOH-H_2_O) to obtain **1** (*t*_R_ = 41.1 min, 10.9 mg) and **5** (*t*_R_ = 50.7 min, 3.2 mg). Fr. D5.3.4 (1049.1 mg) was separated by semipreparative HPLC, using 80% MeOH-H_2_O as the mobile phase, producing the subfraction D5.3.4.3 (155.0 mg), which was subjected to semipreparative HPLC and eluted with 69% CH_3_CN-H_2_O to produce **9** (*t*_R_ = 24.1 min, 11.9 mg).

#### 4.3.1. 2α,3β-Diacetyl Arjunolic Acid (**1**)

White amorphous powder; [*α*]^20^_D_ −33° (c 0.1, MeOH); UV (MeOH) *λ*_max_ (log *ε*): 200 (3.9), 265 (3.2) nm; ^1^H (CDCl_3_, 600 MHz) and ^13^C NMR (CDCl_3_, 150 MHz), see Table 1 and Table 2; HRESIMS [M − H]^−^ *m/z* 571.3633 (calculated for C_34_H_51_O_7_^−^, 571.3640).

#### 4.3.2. 3β,23-Diacetyl Arjunolic Acid (**2**)

White amorphous powder; [*α*]^20^_D_ −2° (c 0.1, MeOH); UV (MeOH) *λ*_max_ (log *ε*): 204 (3.9), 266 (3.2) nm; ^1^H (CDCl_3_, 600 MHz) and ^13^C NMR (CDCl_3_, 150 MHz), see Table 1 and Table 2; HRESIMS [M − H]^−^ *m/z* 571.3639 (calculated for C_34_H_51_O_7_^−^, 571.3640).

#### 4.3.3. 2α-Acetyl Arjunolic Acid (**3**)

White amorphous powder; [*α*]^20^_D_ +6° (c 0.1, MeOH); UV (MeOH) *λ*_max_ (log *ε*): 202 (3.9), 268 (3.2) nm; ^1^H (CD_3_OD, 600 MHz) and ^13^C NMR (CD_3_OD, 150 MHz), see Table 1 and Table 2; HRESIMS [M + H]^+^ *m/z* 531.3694 (calculated for C_32_H_51_O_6_^+^, 531.3680).

#### 4.3.4. 23-Acetyl Hovenic Acid (**4**)

White amorphous powder; [*α*]^20^_D_ −126° (c 0.1, MeOH); UV (MeOH) *λ*_max_ (log *ε*): 207 (3.7), 265 (2.8) nm; ^1^H (CD_3_OD, 600 MHz) and ^13^C NMR (CD_3_OD, 150 MHz), see Table 1 and Table 2; HRESIMS [M − H]^−^ *m/z* 529.3528 (calculated for C_32_H_49_O_6_^−^, 529,3535).

#### 4.3.5. 2α,23-Diacetyl Hovenic Acid (**5**)

White amorphous powder; [*α*]^20^_D_ -22° (c 0.1, MeOH); UV (MeOH) *λ*_max_ (log *ε*): 204 (3.9) nm; ^1^H (CD_3_OD, 600 MHz) and ^13^C NMR (CD_3_OD, 150 MHz), see Table 1 and Table 2; HRESIMS [M − H]^−^ *m/z* 571.3636 (calculated for C_34_H_51_O_7_^−^, 571.3640).

### 4.4. Hydrolysis of 2α,3β-Diacetyl Arjunolic Acid (**1**)

Compound **1** (1.5 mg) was dissolved in methanol, and 1.5 μL of NaOH-H_2_O saturated solution was added and stirred at room temperature for 4 h. Then, the reaction solution was adjusted to pH 5–6 with 10% HCl and filtered to obtain a white solid **1a**.

Arjunolic Acid (**1a**): White amorphous powder; [*α*]^20^_D_ +69° (c 0.1, EtOH); ^1^H NMR (CD_3_OD, 600 MHz) *δ*_H_ 0.70 (3H, s, H-24), 0.82 (3H, s, H-26), 0.91 (3H, s, H-29), 0.95 (3H, s, H-30), 1.03 (3H, s, H-25), 1.18 (3H, s, H-27), 2.86 (1H, dd, *J* = 12.8 and 3.4 Hz, H-18), 3.27 (1H, d, *J* = 9.0 Hz, H-3), 3.35 (1H, d, *J* = 12.0 Hz, H-23a), 3.50 (1H, d, *J* = 12.0 Hz, H-23b), 3.69 (1H, m, H-2), 5.26 (1H, brs, H-5). ^13^C NMR (CD_3_OD, 150 MHz): *δ*_C_ 47.6 (C-1), 69.7 (C-2), 78.2 (C-3), 44.1 (C-4), 48.2 (C-5), 19.1 (C-6), 33.8 (C-7), 40.6 (C-8), 47.9 (C-9), 39.0 (C-10), 24.0 (C-11), 123.4 (C-12), 145.4 (C-13), 43.0 (C-14), 28.8 (C-15), 24.6 (C-16), 47.2 (C-17), 42.7 (C-18), 48.9 (C-19), 31.6 (C-20), 34.9 (C-21), 33.3 (C-22), 66.3 (C-23), 13.9 (C-24), 17.8 (C-25), 17.6 (C-26), 26.5 (C-27), 181.9 (C-28), 33.6 (C-29), 24.0 (C-30).

### 4.5. Cytotoxic Activity

#### 4.5.1. Cell Lines and Tested Compounds

Three kinds of human cancer cell lines (HeLa, MCF-7, HepG2) were cultured in Dulbecco’s Modified Eagle medium (DMEM) (HyClone, Logan, UT, USA) with 10% fetal bovine serum (FBS, ExCell Bio, Shanghai, China) in a humidified atmosphere containing 5% CO_2_ at 37 °C.

The tested compounds were solubilized in DMSO and stored at 4 °C.

#### 4.5.2. Cell Cultures

We determined the anti-cancer effects of compounds **1**–**11** using a CCK8 assay, as reported earlier [56]. The cells were seeded in a 96-well plate as 8 × 10^4^ cells/mL (100 μL/well). In total, 100 μL/well of the serial dilutions of the tested compounds (50, 25, 12.5, 6.25, 3.125 µM) and adriamycin (2.5, 1.25, 0.625, 0.3125, 0.15625 µM) were added to the plate after the overnight incubation of the cells at 37 °C and 5% CO_2_. The cells were incubated for 48 h. Subsequently, 10 μL of CCK8 was added to each well, the plate was incubated for 1 h, and the absorbance of the wells was measured at 450 nm using a Biotech plate reader. Each experiment was repeated three times, and the standard deviation was calculated (±). The concentration that caused a 50% inhibition of cell growth (IC_50_) was calculated for each compound.

## Figures and Tables

**Figure 1 plants-12-00192-f001:**
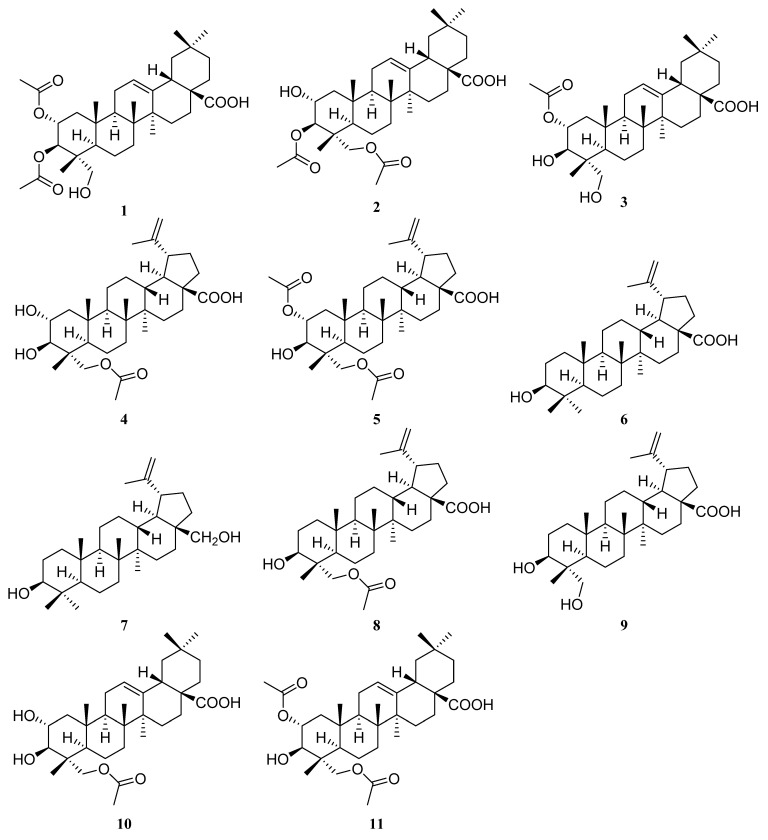
Structures of **1**–**11**.

**Figure 2 plants-12-00192-f002:**
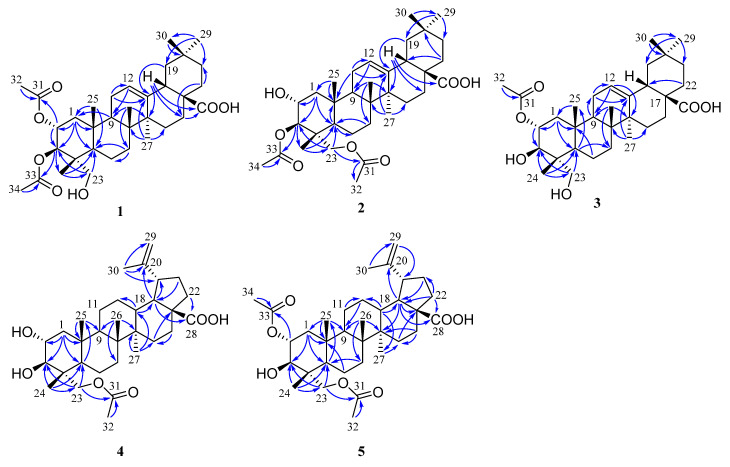
Key HMBC (
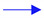
) correlations of **1**–**5**.

**Figure 3 plants-12-00192-f003:**
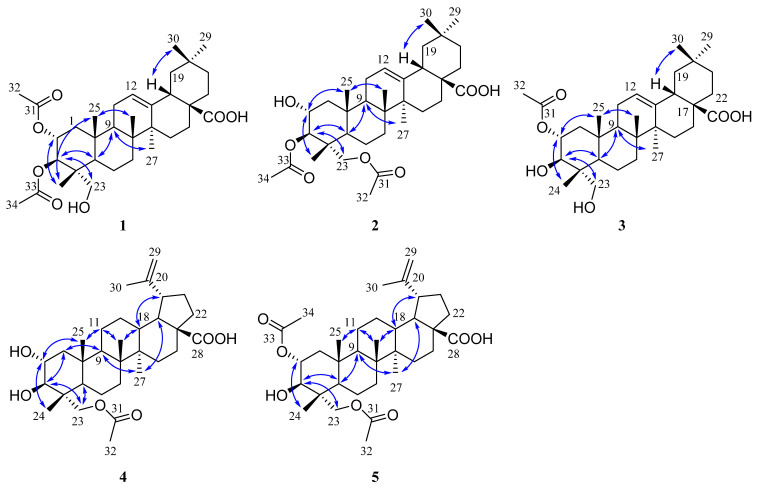
Key NOESY (
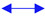
) correlations of **1**–**5**.

**Table 1 plants-12-00192-t001:** ^1^H NMR spectroscopic data of **1**–**6** measured at 600 MHz [*δ*_H_, mult. (*J* in Hz)].

No.	1 ^a^	2 ^a^	3 ^b^	4 ^b^	5 ^b^
1	2.04, dd (12.4, 4.7)	2.07, m	1.96, m	2.02, dd (12.6, 4.8)	2.04, m
	1.02, d (11.5)	1.01, m	0.91, m	0.83, t (12.0)	
2	5.23, td (10.7, 4.9)	3.86, m	5.01, td (10.8, 4.2)	3.68, td (9.7, 4.8)	5.01, td (10.1, 4.2)
3	4.91, d (10.7)	4.80, d (10.1)	3.61, d (10.8)	3.25, d (9.7)	3.50, d (10.1)
5	1.49, m	1.27, m	1.33, m	1.17, m	1.20, d (12.0)
6	1.53, m	1.40, m	1.48, m	1.40, m	1.42, m
7	1.32, m	1.39, m	1.60, m	1.38, m	1.39, m
	1.26, m	1.30, m	1.29, m		
9	1.69, m	1.66, m	1.70, m	1.42, m	1.42, m
11	1.85, m	1.93, m	1.94, m	1.48, m	1.40, m
			1.60, m	1.31, dd (12.9, 4.7)	1.30, dd (12.6, 4.5)
12	5.26, t (3.8)	5.28, t, (3.4)	5.23, br s	1.75, m	1.73, d (13.0)
				1.09, dd (13.2, 4.7)	1.07, m
13				2.33, td (12.7, 3.1)	2.34, t (11.4)
15	1.70, m	1.70, m	1.80, m	1.54, td (13.1, 3.3)	1.55, td (13.2, 4.1)
	1.07, m	1.05, m	1.07, m	1.17, m	1.16, m
16	1.96, m	1.93, m	1.96, m	2.24, dt (12.6, 2.7)	2.24, dt (12.8, 2.8)
	1.61, m	1.61, m	1.88, m	1.42, m	1.11, m
18	2.81, dd (13.8, 3.7)	2.82, dd (13.4, 3.9)	2.87, d (12.7)	1.62, t (11.5)	1.62, t (11.3)
19	1.59, m	1.61, m	1.66, m	3.03, td (10.8, 4.6)	3.03, m
	1.14, m	1.17, m	1.11, m		
21	1.34, m	1.34, m	1.38, m	1.92, m	1.92, m
	1.21, m	1.21, m	1.20, m	1.40, m	1.40, m
22	1.76, m	1.76, m	1.74, m	1.91, m	1.90, m
	1.55, m	1.56, m	1.52, m	1.45, m	1.45, m
23	3.37, d (12.5)	3.82, d (12.1)	3.51, d (10.8)	3.93, s	3.95, s
	2.89, d (12.7)	3.65, d (12.0)	3.28, d (11.1)		
24	0.72, s	0.84, s	0.73, s	0.75, s	0.80, s
25	1.08, s	1.01, s	1.08, s	0.96, s	1.01, s
26	0.74, s	0.74, s	0.84, s	0.97, s	0.97, s
27	1.13, s	1.12, s	1.17, s	1.01, s	1.01, s
29	0.90, s	0.90, s	0.90, s	4.72, d (2.2)	4.71, s
				4.60, br s	4.59, s
30	0.92, s	0.92, s	0.94, s	1.70, s	1.70, s
32	2.08, s	2.08, s	2.05, s	2.06, s	2.07, s
34	2.00, s	2.11, s			2.04, s

^a^ Measured in CDCl_3_. ^b^ Measured in CD_3_OD.

**Table 2 plants-12-00192-t002:** ^13^C NMR spectroscopic data of **1**–**5** measured at 125 MHz.

No.	1 ^a^	2 ^a^	3 ^b^	4 ^b^	5 ^b^	No.	1 ^a^	2 ^a^	3 ^b^	4 ^b^	5 ^b^
1	43.9	47.5	44.7	48.2	45.0	18	41.0	41.0	42.9	50.4	50.4
2	69.6	67.6	74.2	69.5	73.9	19	46.0	45.9	47.5	48.5	48.5
3	76.3	79.0	74.6	77.7	74.5	20	30.8	30.8	31.7	152.0	152.0
4	43.7	42.1	44.6	43.5	43.9	21	33.9	33.9	35.0	31.7	31.8
5	46.4	47.8	47.9	49.3	49.6	22	32.2	32.5	34.0	38.2	38.2
6	17.8	18.1	19.0	19.2	19.2	23	64.6	65.5	65.6	67.0	66.7
7	32.5	32.3	33.3	35.2	35.1	24	13.8	14.0	13.8	13.6	13.5
8	39.5	39.4	40.6	42.0	42.0	25	17.2	17.1	17.3	18.3	18.1
9	47.6	47.9	49.6	52.1	52.0	26	17.2	17.2	17.8	16.6	16.6
10	38.2	38.1	39.2	39.2	39.6	27	26.1	26.0	26.5	15.0	15.0
11	23.6	23.6	24.2	22.2	22.3	28	183.6	183.9	-c	180.2	180.3
12	122.2	122.4	123.1	26.8	26.8	29	33.2	33.2	33.6	110.2	110.2
13	144.0	143.8	145.7	39.6	39.4	30	23.7	23.7	24.1	19.5	19.6
14	41.8	41.7	43.1	43.6	43.6	31	170.6	171.0	173.0	172.7	172.6
15	27.7	27.7	28.9	30.8	30.8	32	21.3	21.0	21.3	20.8	20.8
16	23.0	22.9	24.6	33.4	33.4	33	172.8	172.3			172.9
17	46.6	46.6	30.8	57.5	57.6	34	20.9	21.2			21.3

^a^ Measured in CDCl_3_. ^b^ Measured in CD_3_OD. ^c^ Not observed.

**Table 3 plants-12-00192-t003:** Cytotoxic activities of compounds **1**–**11** against three cancer cell lines.

Compound	Cell lines and IC_50_ (μM)		
	HeLa	HepG-2	MCF-7
**1**	17.35 ± 0.58	16.76 ± 0.95	15.12 ± 1.06
**2**	16.96 ± 1.24	13.77 ± 1.15	21.55 ± 0.59
**3**	>50	>50	>50
**4**	26.06 ± 1.72	24.18 ± 1.01	>50
**5**	19.62 ± 1.92	>50	13.22 ± 1.24
**6**	>50	>50	>50
**7**	>50	>50	>50
**8**	5.04 ± 0.15	9.76 ± 0.96	8.94 ± 0.70
**9**	13.88 ± 0.21	>50	13.46 ± 1.26
**10**	40.58 ± 0.36	35.24 ± 0.88	>50
**11**	17.51 ± 0.73	12.80 ± 1.04	13.27 ± 0.42
Doxorubicin	2.19 ± 0.08	1.72 ± 0.19	1.55 ± 0.09

## Data Availability

The data presented in this study are available in the Appendix A.

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
