# Peer review of "Triterpene Derivatives from Garcinia oligantha and Their Anti-Cancer Activity"

_plants, 2023, doi:10.3390/plants12010192_

Round 1
Reviewer 1 Report
Please see the attachment.

Author Response
December 26, 2022
Plants
Manuscript ID: plants-2083759
Title: “Triterpene Derivatives from Garcinia oligantha and Their Anticancer Activity”
Author(s): Xiao-Hui Peng, Chao Wang, Yong-Lian Hou, Jia-Mei Tian, Xiao-Jie Fan, Da-Hong Li, Hui-Ming Hua
Dear reviewer,
Thank you for giving us the opportunity and valuable suggestions to improve the quality of our manuscript. Your comments are laid out below in bold font and our response is given in normal font and changes to the manuscript are marked up using the “Track Changes” function in the manuscript. We hope these responses are satisfactory, and please let us know if any additional data and/or information is needed.
Best regards,
Sincerely yours,
Huiming Hua, Prof. Dr.
School of Traditional Chinese Materia Medica
Shenyang Pharmaceutical University
People’s Republic of China
E-mail: hmhua@syphu.edu.cn
Response to the comments
- What’s the configuration of the hydrogen at C-3 in compound 8?
Our response:
It’s beta orientation.
- For the red part of the manuscript “The 1H NMR spectrum (Table 1) of 1 displayed signals corresponding to six tertiary methyls at δH 1.13 (3H, s), 1.08 (3H, s), 0.92 (3H, s), 0.90 (3H, s), 0.74 (3H, s), and 0.72 (3H, s), and two acetyl groups at δH 2.08 (3H, s) and 2.00 (3H, s), as well as an olefinic proton at δH 5.26 (1H, t, J = 3.8 Hz)”. The reviewer thinks if it is reported in the experimental section it will be enough. Otherwise the manuscript becomes unwieldy to read.
Our response:
These methyl and olefinic proton signals are the characteristic and highly recognizable signals for oleanane-type triterpene. So, in many journals, including Plants, these characteristic signals were described in the process of structure elucidation.
- Please place the structures in Figure 1 before the data tables.
Our response:
We moved Figures 1-3 to the front of Table 1.
- Please reduce the size of the numbering of the carbon atoms in these figures.
Our response:
We reduced the size of these serial numbers in Figures 2 and 3.
- Make all your structures the same size as for Figure 3. The quality of the figures needs to be improved.
Our response:
We unified the size of these compounds, and improved the quality of these pictures.
- Please have the table headers repeat when the table goes overleaf. Makes for much easier reading. Either that, or reduce the font size so the table appears on a single page.
Our response:
After adjusting the image position, each table can be completely placed in a single page.
- For the red part of the manuscript “The 1H and 13C NMR data of 2 and 1 showed close similarity with the major differences on the chemical shifts of C-2, and H-2 upfield shifted from δC 69.57, and δH 5.23 to δC 67.63, and δH 3.86, and C-23 and H2-23 downfield shifted from δC 64.57, and δH 3.37, 2.89 to δC 65.46, and δH 3.82, 3.65, respectively, suggested that the acetoxy at C-2 in 1 was shifted to C-23 in 2.” The reviewer thinks its is not logically consistent, since that 1H was put before 13C, but in the text discussing the 13C data was put before 1H data. Similarly, structure 2 should not be mentioned before structure 1.
Our response:
We change the sentence to " The 13C and 1H NMR data of 1 and 2 showed…… ".
- Paar?
Our response:
We changed the initial letter to uppercase according to your requirements.
- Why was this fraction (D4.3.1) chosen? Because of its bulk?
Our response:
The large quantity is indeed one of the reasons for choosing it.
- This is not usually called a residue. Usually an extract. However, you extract this material further. Perhaps call it a resin? What was the nature of this material before it was extracted with the series of solvents?
Our response:
We replaced the word “residue” with the word “resin”. It was black sold before it was extracted with the series of solvents.
- For the red part of the manuscript “Akongwi, M.; Tih, A.E.; Nyongbela, K.D.; Samje, M.; Ghogomu, R.T. Brevipedicelones D and E, two C-O-C flavonoid dimmers from the leaves of Garcinia brevipedicellata and anti-onchocercal activity. Nat. Prod. Bioprospect. 2019, 9, 61-68.” Was the original paper incorrect in referring to dimmers?
Our response:
We checked the journal and found it is “dimmer” in the original paper, “dimmer” should be “dimer”.
- For the part of the manuscript “Compound 2 displayed an [M – H] – ion peak at m/z 571.3639 (calcd. 571.3640) in its negative mode HRESIMS, corresponding to the molecular formula C34H52O7”. Why is this text indented?
Our response:
It is the requirement of the journal.
- Why the activity of compound 12 was not tested.
Our response:
Compound 12 was obtained two years ago, of which the structure has been determined by its NMR data. Unfortunately, the sample of 12 was lost during the lab moving, so the cytotoxicity of compound 12 was not tested. According to the opinion of another reviewer, we have deleted compound 12.

Reviewer 2 Report
The authors describe the identification of five new compounds but with limited novelty since are acetylated derivatives (mono or diacetylated) of known compounds.
The paper is well presented and the desciption of new structures is consistent.
There are only some modifications needed.
Please refer to the observations/suggestions available as notes in the enclosed file.

Author Response
- In this context several papers should be also considered.
Our response:
We have added some references you recommended in the introduction section.
- Generally the chemical shifts of carbons are reported with only one decimal digit.
Our response:
We changed all carbon spectroscopic data in the full text to one decimal digit.
- It would be easier to follow this fractionation by a scheme that the authors may provide as supplementary materials.
Our response:
We draw a separation flow chart as shown in the figure below and put it in the Supporting information (Figure S82). High resolution original pictures will be submitted in the attachment.
- This is the formula of one ion and should be provided with the corresponding electric charge.
Since it is one deprotonated species its formula should be as C32H49O6-, Provided with one negative charge.
Our response:
We have modified the format according to your suggestions.
Reviewer 3 Report
Journal: Plants
Manuscript ID: plants-2083759
Type of manuscript: Article
Title: Triterpene derivatives from Garcinia oligantha and their anticancer activity
Authors: Xiao-Hui Peng, Chao Wang, Yong-Lian Hou, Jia-Mei Tian, Xiao-Jie Fan, Da-Hong Li, Hui-Ming Hua
In the present study, five new and seven known triterpenoids were identified from the leaves and twigs of Garcinia oligantha, of which 11 isolates were tested for their cytotoxicity against human MCF7, MDA-MB-231, and HS578T breast cancer cells, and some compounds were active. However, the configuration determination for all the new compounds is not clear, for which the authors need to hydrolyze compounds 1 and 4 and compare the analytic data of the products with those of arjunolic acid and hovenic acid, respectively. In addition, the authors should discuss why they did not isolate xanthones from Garcinia oligantha, as reported previously, and why the cytotoxicity of compound 12 was not tested. Thus, this manuscript could be re-considered as a potential publication in Plants after the authors address these issues in their revision of the present manuscript.
Author Response
- The configuration determination for all the new compounds is not clear, for which the authors need to hydrolyze compounds 1 and 4 and compare the analytic data of the products with those of arjunolic acid and hovenic acid, respectively.
Our response:
Thanks for your suggestions. We hydrolyzed compound 1 and obtained the hydrolysate 1a. The 1D NMR data and specific rotation of 1a was tested, and 1a was determined as arjunotic acid. Therefore, the configuration of 1 and related compounds was finally determined. The description of the hydrolysis experiment of compound 1 and NMR data of the hydrolysate 1a were supplemented to the text, and the NMR spectra of 1a was added in SI. Unfortunately, the remaining amount of compounds 4 and 5 is particularly small., which is not sufficient for hydrolysis. However, considering that the relative configuration of the same type of triterpenoids reported in many literatures was determined by coupling constants and NOESY spectral data, and combining the biogenetic relationship, we think that the configuration of 4 determined by the coupling constants and NOESY data is acceptable.
- The authors should discuss why they did not isolate xanthones from Garcinia oligantha, as reported previously.
Our response:
We have indeed isolated some known xanthones. Since there are very few reports of triterpenoids in the genus Garcinia, this paper focuses on the triterpenoids from the G.oligantha to enrich the reports on the natural products of triterpenoids in the genus and antitumor activity.
- Why the cytotoxicity of compound 12 was not tested.
Our response:
Compound 12 was obtained two years ago, of which the structure has been determined by its NMR data. Unfortunately, the sample of 12 was lost during the lab moving, so the cytotoxicity of compound 12 was not tested.
Round 2
Reviewer 3 Report
Journal: Plants
Manuscript ID: plants-2083759
Type of manuscript: Article
Title: Triterpene derivatives from Garcinia oligantha and their anticancer activity
Authors: Xiao-Hui Peng, Chao Wang, Yong-Lian Hou, Jia-Mei Tian, Xiao-Jie Fan, Da-Hong Li, Hui-Ming Hua
In the present study, five new and seven known triterpenoids were identified from the leaves and twigs of Garcinia oligantha, of which 11 isolates were tested for their cytotoxicity against human MCF7, MDA-MB-231, and HS578T breast cancer cells, and some compounds were active. However, the configuration determination for all the new compounds is not clear, for which the authors need to discuss their hydrolysis results and comparing the specific rotation value of the hydrolyzed product of 1 with that of arjunolic acid, for which the authors also need to compare the NMR spectroscopic data. Similarly, the authors need to compare both NOESY NMR spectra and the specific rotation values of 4 with those of hovenic acid to support the configuration determination. In addition, compound 12 may not be reported if the sample was not available for the future reference, but compound 1a could be included in the manuscript. Thus, this manuscript could be published in Plants after the authors address these issues in their further revision of the present manuscript.
Author Response
December 21, 2022
Plants
Manuscript ID: plants-2083759
Title: “Triterpene Derivatives from Garcinia oligantha and Their Anticancer Activity”
Author(s): Xiao-Hui Peng, Chao Wang, Yong-Lian Hou, Jia-Mei Tian, Xiao-Jie Fan, Da-Hong Li, Hui-Ming Hua
Dear editors,
Thank you for giving us the opportunity and valuable suggestions to improve the quality of our manuscript. We have carefully responded all the reviewers’ comments as itemized below. The reviewers’ comments are laid out below in bold font and our response is given in normal font and changes to the manuscript are marked up using the “Track Changes” function in the manuscript. We hope these responses are satisfactory, and please let us know if any additional data and/or information is needed.
Best regards,
Sincerely yours,
Huiming Hua, Prof. Dr.
School of Traditional Chinese Materia Medica
Shenyang Pharmaceutical University
People’s Republic of China
E-mail: hmhua@syphu.edu.cn
Response to the comments
Reviewer #2:
- The configuration determination for all the new compounds is not clear, for which the authors need to discuss their hydrolysis results and comparing the specific rotation value of the hydrolyzed product of 1 with that of arjunolic acid, for which the authors also need to compare the NMR spectroscopic data.
Our response:
The specific rotation value of the hydrolyzed product of 1 was [α]20D +69° (c 0.1, EtOH), which was similar to that of arjunolic acid [α]19D +63.5° (c 0.51, EtOH). We also afforded the NMR spectroscopic data in the section of “4.4. Hydrolysis of 2α,3β-Diacetyl Arjunolic Acid (1)”, which were same as those reported [51].
- The authors need to compare both NOESY NMR spectra and the specific rotation values of 4 with those of hovenic acid to support the configuration determination.
Our response:
The specific rotation value of the hydrolyzed product of 4 was [α]20D -126° (c 0.1, MeOH), which was similar to that of hovenic acid [α]25D -30.3° (c 0.2, MeOH). The NOESY NMR spectra data of these two compounds were also similar.
- Compound 12 may not be reported if the sample was not available for the future reference, but compound 1a could be included in the manuscript.
Our response:
Following your suggestion, we deleted the report of compound 12 and compound 1a was included in the manuscript.
